# Potentially inappropriate prescriptions according to explicit and implicit criteria in patients with multimorbidity and polypharmacy. MULTIPAP: A cross-sectional study

**Juan A. Lopez-Rodriguez**[1,2,3,4©], **Eloísa Rogero-Blanco**[1,2,4©]*, **Mercedes Aza-Pascual-Salcedo**[5], **Fernando Lopez-Verde**[6,7], **Victoria Pico-Soler**[4,8,9], **Francisca Leiva-Fernandez**[6,10], **J. Daniel Prados-Torres**[4,6,10‡], **Alexandra Prados-Torres**[4,8,11‡], **Isabel Cura-González**[2,3,4‡], **MULTIPAP group**¶

1 Primary Healthcare Center General Ricardos, Madrid, Spain, 2 Medical Specialties and Public Health Department, School of Health Sciences, University Rey Juan Carlos Alcorcón, Madrid, Spain, 3 Research Support Unit, Primary Care Management, Madrid, Spain, 4 Health Services Research on Chronic Patients Network (REDISSEC), Madrid, Spain, 5 Primary Care Direction, Sector Zaragoza III, Zaragoza, Spain, 6 Unidad Docente Multiprofesional de Atención Familiar y Comunitaria Distrito Málaga/Guadalhorce, Málaga, Spain, 7 Primary Healthcare Center Las Delicias, Málaga, Spain, 8 EpiChron Research Group, IIS Aragón, Aragon Health Sciences Institute, Zaragoza, Spain, 9 Primary Healthcare Center Torrero-La Paz, Zaragoza, Spain, 10 Biomedical Research Institute of Malaga (IBIMA), Malaga, Spain, 11 Miguel Servet University Hospital, Zaragoza, Spain

☉ These authors contributed equally to this work.
‡ JDPT, APT and ICG are Joint Senior Authors.
¶ Membership of the MULTIPAP group is provided in the Acknowledgments.
* mariaeloisa.rogero@salud.madrid.org

George OPEN ACCESS

**Data Availability Statement:** Regarding data exchange, the Aragon Ethics Committee approved

## Abstract

### Background

Multimorbidity is a global health challenge that is associated with polypharmacy, increasing the risk of potentially inappropriate prescribing (PIP). There are tools to improve prescription, such as implicit and explicit criteria.

### Objective

To estimate the prevalence of PIP in a population aged 65 to 74 years with multimorbidity and polypharmacy, according to American Geriatrics Society Beers Criteria® (2015, 2019), the Screening Tool of Older Person's Prescription -STOPP- criteria (2008, 2014), and the Medication Appropriateness Index -MAI- criteria in primary care.

### Methods

This was an observational, descriptive, cross-sectional study. The sample included 593 community-dwelling elderly aged 65 to 74 years, with multimorbidity and polypharmacy, who participated in the MULTIPAP trial. Socio-demographic, clinical, professional, and

this research without considering the option of data sharing. The data contains sensitive clinical information about the patient, so there are ethical and legal restrictions to sharing the data set. The data are part of the MULTIPAP study and can be requested by contacting the Aragon Ethics Committee at the email address ceica@aragon.es; for the request of data you can also contact the Primary Care Management of Madrid at the email address gap@salud.madrid.org; and by contacting the Technical Direction of Teaching and Research at the email address dtdei@salud.madrid.org The MULTIPAP Group may establish future collaborations with other groups based on the same data. The main researchers of the project will be contacted (Alexandra Prados-Torres at sprados.iacs@aragon.es; Daniel Prados-Torres at uand.prados.sspa@juntadeandalucia.es; and Isabel del Cura at isabel.cura@salud.madrid.org). However, each new project based on these data must be previously submitted to CEICA for approval.

**Funding:** This study was funded by National Institute for Health Research ISCIII (Grant numbers PI15/00276 (APT), PI15/00572 (ICG), PI15/00996 (JDPT), RD16/0001/0004 (ICG), RD16/0001/0005 (APT), RD16/0001/0006 (JDPT)) Co-funded by European Regional Development Fund, (ERDF) "A way of shaping Europe". National Plan I+D+I 2013-2016". ERB has received a grant from the Foundation for Biomedical Research and Innovation in Primary Care (FIIBAP) for translation in its 2019 call and received funding from the Spanish Society of Family and Community Medicine -semFYC- as it won a grant for the completion of doctoral theses Isabel Fernández 2018.

**Competing interests:** The authors have declared that no competing interests exist.

pharmacological-treatment variables were recorded. Potentially inappropriate prescribing was detected by computerized prescription assistance system, and family doctors evaluated the MAI. The MAI-associated factors were analysed using a logistic regression model.

## Results

A total of 4,386 prescriptions were evaluated. The mean number of drugs was 7.4 (2.4 SD). A total of 94.1% of the patients in the study had at least one criterion for drug inappropriateness according to the MAI. Potentially inappropriate prescribing was detected in 57.7%, 43.6%, 68.8% and 71% of 50 patients according to the explicit criteria STOPP 2014, STOPP 2008, Beers 2019 and Beers 2015 respectively. For every new drug taken by a patient, the MAI score increased by 2.41 (95% CI 1.46; 3.35) points. Diabetes, ischaemic heart disease and asthma were independently associated with lower summated MAI scores.

## Conclusions

The prevalence of potentially inappropriate prescribing detected in the sample was high and in agreement with previous literature for populations with multimorbidity and polypharmacy. The MAI criteria detected greater inappropriateness than did the explicit criteria, but their application was more complex and difficult to automate.

## Introduction

Multimorbidity, defined by the WHO as the coexistence of two or more chronic illnesses in a person [1], is a growing phenomenon. It describes the complex interaction in a patient of several co-existing diseases. It has become a health problem and an international health challenge [2] resulting from increased life expectancy and non-transmittable disease rates, among other factors.

Patients with multimorbidity usually present polypharmacy, defined as the simultaneous use of several medicines[3]. Data from the USA National Health Survey yielded a prevalence of polypharmacy of 39% in the population aged over 65 years [4]. An European study, with medication dispensing data for 310,000 adults, observed that the proportion of patients with ≥5 drugs dispensed doubled, reaching 20.8% in the 1995–2014 period[5]. In the National Health Survey in Spain in 2017, in this same age range, 27.7% of respondents reported consuming at least 5 or more drugs or pharmaceutical preparations [6].

Polypharmacy entails a greater risk for potentially inappropriate prescribing (PIP), which is defined as "the prescribing of medications that should generally be avoided in persons 65 years or older because they are either ineffective or they pose unnecessarily high risk where a safer alternative is available" [7]. Optimization and the appropriateness of prescriptions in this population has become a global public health problem.

To analyse potentially inappropriate prescriptions in primary care complex patients, explicit and implicit methods have been proposed [8,9]. Explicit methods, are focused on drugs (e.g., Beers and STOPP/START criteria), measuring how they fit to a set of predefined criteria [10–12]. These criteria are updated according to the available evidence for each different drug catalogues (USA or Europe). The implicit criteria are based on the global assessment by a health professional (pharmacist, internist, geriatrician or primary care physician), which takes into account the overall situation of the patient and whether the prescription

corresponds to an indication or need. The most accepted implicit method internationally is the medication appropriateness index (MAI) [13].

Depending in the criteria used, the prevalence of patients with PIP differs significantly depending on the tools and the populations analysed (hospitalized, institutionalized or community patients) [14,15]. It ranges from 40.4% detected using STOPP criteria [16] to 98.7% of prevalence using MAI criteria in primary care young elderly patients taking at least 5 drugs [17]. In Spain, the MAI has been evaluated in hospitalised patients [18] and has also been adapted and validated for primary care [19]. However, we still lack in young-elderly population studies in primary care patients complaining with multimorbidity and polypharmacy.

The association between explicit methods (STOPP or Beers) and implicit methods (MAI) in the primary care population has been investigated [20]. Recently, a study was published comparing the sensitivity and specificity of these three methods performed by two pharmacists, using the MAI as the *gold standard*, in a population from Kuwait. This study compared Beers 2015, STOPP 2014 and FORTA with the MAI, obtaining *kappa values* of 0.16 (p <0.001), 0.40 (p <0.001) and 0.23 (p <0.001), respectively [21]. The most updated version of Beers 2019 has not been compared with the rest of the criteria.

Studies have shown association of high prevalence of PIP with worse health outcomes in patients such as unscheduled outpatient visits, emergency room visits, adverse events [17,22–24], and hospital admissions and re-admissions [25–28]. Furthermore, this association has also been found when analysing patient reported outcomes (PROMs), such as quality of life (EQ-5D-3L) [29].

In an attempt to facilitate the detection of PIP in patients, in recent years, explicit criteria (STOPP and Beers) have been automated into computerized prescription assistance systems (CPAS) [30]. The translation of explicit criteria to computer algorithms is complex [31,32] although easier than programming the implicit criteria (MAI) to computer pre-sets. The MAI criteria imply evaluating ten different aspects of each drug including indication, dose, effectiveness, interactions and duration, among others. Some of this evaluations even require health professional confirmation making this implicit criteria time consuming [13,33–35].

The main aim of this study was to estimate the prevalence of PIP in a population aged 65 to 74 years with multimorbidity and polypharmacy, according to explicit criteria (Beers 2015 and 2019, STOPP 2008 and 2014); and implicit criteria (Medication appropriateness index–MAI-) in primary care. The secondary objectives were to assess the clinical and socio-demographic predictors for PIP and to compare these criteria in order to detect PIP with greater applicability to the Spanish population.

## Material and methods

The project has been favourably evaluated by the Central Committee of Primary Care Research of the Community of Madrid, the Commission for Health Research of the Aragon Institute for Health Research (IIS Aragon), and the Commission for Health Research of the Bio-Sanitary Research Institute in Malaga (IBIMA). The trial protocol was approved by the Clinical Research Ethics Committee of Aragon (CEICA) on September 30, 2015 (CP14/2015), and by the Research Ethics Committee of the Province of Malaga on September 25, 2015. Participant provided written consent to participate in the Multi-PAP trial Clinicaltrials.gov NCT02866799.

### Design

This was a cross-sectional, descriptive, multicentric, observational study conducted in the Spanish primary care setting.

## Participants and setting

Patients aged 65 to 74 years with multimorbidity ($\geq$ 3 diseases) and polypharmacy ($\geq$ 5 prescriptions taken for at least 3 months), who had attended their doctor consultation at least once over the last year and provided written consent to participate in the Multi-PAP trial Clinicaltrials.gov NCT02866799 were included [36]. Institutionalized patients, those whose life expectancy was <12 months, as estimated by their doctor, and patients with any severe mental disorder were excluded. For physicians, the inclusion criterion was at least one year in their job. Presentations of the Multi-PAP project were held in health centres, and professionals were offered the chance to participate. One hundred and seventeen family doctors from 38 healthcare centres from three Spanish regions (Andalucia, Aragon, and Madrid) and 593 patients agreed to participate. For the proposed objective and based on previous studies in a similar population reporting an average MAI summated score of 15.8 (10.1 SD) [17], with this sample size and a design effect of 1.2, a maximum type I error of 0.975% with a 95% confidence interval (95%CI) was determined.

## Data collection

Sociodemographic and clinical data were obtained during the period of patient recruitment in the MULTIPAP study, which was conducted between December 2016 and January 2017 through an interview with each patient's family doctor.

The method of collecting data on patients and doctors was not the same. Data on professionals (age, sex, length of physician career, and being postgraduate medical supervisor) were recorded by each physician in the data collection notebook after signing their commitment to collaborate in the study. Subsequently, sociodemographic and clinical data on patients were collected through an interview with each patient's general practitioner after signing the written informed consent.

Subsequently, the data were uploaded from the DCN into the CPAS ChecktheMeds®.

## Variables

The following socio-demographic variables were recorded for patients: age, sex, marital status, education level, social class according to the Spanish classification [37], and family income in thousands of euros adjusted by the number of people in the household (using the method proposed by the OECD Organisation for Economic Co-operation and Development). Additionally, the following clinical variables were collected: number of active pharmaceutical ingredients per patient according to the Anatomical, Therapeutic, Chemical (ATC) classification system; and chronic conditions in accordance with the international classification of primary care (ICPC), with the most relevant ones selected according to the criterion by O'Halloran [38]. The variables for the professional were age, sex, length of physician career, and being postgraduate medical supervisor.

## Evaluation of prescribing appropriateness

One researcher with broad clinical and therapeutic drug monitoring expertise supervised information transfer to the ChecktheMeds® tool and used this tool to globally review the treatment of all patients. ChecktheMeds® is a web-based tool to help health professionals for treatment plan review. It quickly analyses potentially inappropriate prescriptions, potentially drug-drug and drug-disease interactions, contraindications and adverse events. It takes into account clinical variables, diagnosis codes and the whole treatment of a patient. All collected data were

supervised by a second reviewer. PIP was identified using the four explicit criteria selected and using one implicit criterion.

The explicit criteria were a) the 2019 Updated Beers criteria by the American Geriatric Society [11], b) the 2015 Beers criteria by the American Geriatrics Society [39], c) the 2014 v2 STOPP criteria [12], and d) the 2008 v1 STOPP criteria [40]. All the STOPP and Beers criteria were analysed. In agreement with former studies and to avoid potential information bias, this research team agreed on omitting the A1P STOPP criterion from the analysis (any drug prescribed but not indicated by clinical evidence) to prevent its overestimation [41]. START criteria assessing potential prescribing omissions were dismissed because the focus of this study was on already-prescribed inappropriate medications, instead of the need of starting new medications. The four criteria were automatically reported by the CPAS and compared.

The implicit criterion was the MAI [13], which includes 10 implicit criteria (indication, effectiveness, dosage, correct directions, practical directions, drug-drug interactions, drug-disease interaction, duplication, duration and expense). The MAI measures the suitability of each of a patient's drugs to these 10 criteria on a 3-grade Likert scale (A appropriate, C inappropriate). Only the "correct direction" criterion was omitted because this information was not able to be evaluated.

Each medication prescribed was rated by an independent family physician with pharmacological expertise for each Region, applying the usual weighted criteria (three raters in total). The raters were able to review the clinical history of the patients in the study to obtain as much information as possible. The drug-drug and drug-disease interactions were detected with the help of the CPAS ChecktheMeds®. A pilot rating of 17 different drugs (three patients) was carried out between the three raters using the same electronic health records and cases. Discrepancies were discussed until a consensus was reached. Additionally, a family physician and a pharmacist conducted a second appraisal of the inter-observer reliability over a randomly selected 10% of the completed questionnaires.

The MAI was evaluated by drug and by patient. The following were calculated: the percentage of drugs with at least one inappropriate criterion; the percentage of patients with at least one inappropriate criterion for any of the drugs [13]; the percentage of patients with a summated MAI of 3 or more [20]; and the summated MAI score per drug and per patient.

## Statistical analysis

Categorical variables are presented as frequencies and percentages. Quantitative variables are presented as the mean and standard deviation, with the corresponding CI 95% when the data fit a normal distribution or by the median and interquartile range (IQR) in the case of asymmetric data distribution. The MAI was used a reference point due to it's reliably and validity as a standardized assessment tool. For the gold standard test (MAI), a pilot rating for the level of absolute agreement across the three raters was evaluated using the intraclass correlation coefficient (ICC). Specificity and sensitivity of the explicit criteria were assessed using a contingency table, and confidence intervals were calculated using exact correction. Concordance between implicit and explicit criteria was estimated using kappa statistics. A multivariate linear regression model was developed to assess factors associated with the summated MAI score per patient (maximum of 18 points multiplied by each drug prescribed). The independent variables were those reaching statistical significance in the univariate analysis or considered of clinical relevance. Five different multivariate logistic regression models were built to determined factors that were independently associated with PIP, with robust estimators that controlled for the effect of cluster sampling. For each model, the dependent variable was constructed as follows: (I) being prescribed at least one PIP according to Beers 2019 criteria

(0 = no, 1 = yes); (II) being prescribed at least one PIP according to Beers 2015 criteria (0 = no, 1 = yes); (III) being prescribed at least one PIP according to STOPP 2014 criteria (0 = no, 1 = yes); (IV) being prescribed at least one PIP according to STOPP 2008 criteria (0 = no, 1 = yes); and (V) being prescribed medication with one or more inappropriate rating in the MAI criteria (0 = summated patient score 0, 1 = summated patient score≥1) according to the original Hanlon definition [13]. Stata v14.0 software was employed for the statistical analyses.

## Ethical approval

The MULTIPAP study was designed in accordance with the basic ethical principles of autonomy, beneficence, justice, and non-maleficence and was conducted in accordance with the rules of Good Clinical Practice outlined in the most recent Declaration of Helsinki and the Oviedo Convention (1997). Written informed consent of patients was required. Data confidentiality and anonymity was ensured, according to the provisions of Spanish Law 15/1999, both during the implementation phase of the project and in any resulting presentations or publications. The project has been favourably evaluated by the Central Committee of Primary Care Research of the Community of Madrid, the Commission for Health Research of the Aragon Institute for Health Research (IIS Aragon), and the Commission for Health Research of the Bio-Sanitary Research Institute in Malaga (IBIMA). It was approved by the Clinical Research Ethics Committee of Aragon (CEICA) on September 30, 2015, and by the Research Ethics Committee of the Province of Malaga on September 25, 2015.

## Results

### Characteristics of the study participants

A total of 4,386 prescriptions were rated for the 593 included patients. The mean age of the patients was 69.7 (2.7 SD) years, the age range was 65–74, 55.8% were women, 75.4% were married, and 17.9% lived alone. Table 1 provides the main socio-demographic and clinical characteristics of the patients and the professionals variables. Among chronic clinical conditions, 78.9% of patients had hypertension, and 50.8% hypercholesterolemia. The mean number of chronic conditions and medications per patient were 5.8 (2.3 SD) diseases and 7.4 (2.4 SD) prescriptions per patient. 17.9% of patients were prescribed ≥10 drugs (Table 1).

For ATC groups, one of every three drugs (34.1%) prescribed belonged to the cardiovascular group, followed by metabolism and alimentary tract group and nervous system group. Table 2 shows the categories of drugs prescribed to patients according to ATC classification.

### Inappropriate prescribing ratings

The implicit criteria, the MAI, was evaluated for 4,386 prescriptions (589 patients). Four patients could not be evaluated due to not being able to access their medical records at the time of the evaluation. Moderate absolute agreement between the three raters was found, with an intraclass correlation coefficient of 0.41 (CI 95% 0.109–0.701). More than half of the prescriptions were considered appropriate (51.9%). The mean summated MAI score per drug was 1.4 (2.3 SD) (median 0; IQR 0–1). The mean summated MAI score per patient was 17.5 (16.8 SD) (median 14, IQR 5–25; range 0–102). Five hundred and fifty-four patients out of 589 had one or more inappropriate rating among their prescribed medications (94.1%). Of the 2,110 drugs considered inappropriate, 1,416 (67.1%) had less than three MAI inappropriate criteria ratings according to the Steinman classification [20], and 694 (32.9) met three or more MAI inappropriate criteria ratings. The MAI criteria with the highest inappropriate percentages were cost-effectiveness, duration, effectiveness and potential drug-drug interactions. Table 3

**Table 1. Socio-demographic, clinical, and pharmacological characteristics of patients and physicians characteristic.**

| Patient Characteristics (N = 593) | |
|---|---|
| Age, M (SD) | 69.7 (2.7) |
| Gender, n (%) | |
| Female | 331 (55.8) |
| Male | 262 (44.2) |
| Nationality, n (%) | |
| Spanish | 583 (98.3) |
| Other | 10 (1.7) |
| Marital status, n (%) | |
| Single, divorced, widow | 146 (24.6) |
| Married, living with couple | 447 (75.4) |
| Living alone, n (%) | 106 (17.9) |
| Educational level, n (%) | |
| Primary education not completed | 279 (47.0) |
| Primary education | 196 (33.1) |
| Secondary and superior education | 118 (19.9) |
| Social class according to occupation, n (%) | |
| Supervisors, managers, and directors | 234 (39.5) |
| Skilled primary sector | 217 (36.6) |
| Unskilled | 142 (23.9) |
| **Patient Clinical Conditions** | |
| Number of chronic illnesses*, median (IQR) | 5.0 (4.0, 7.0) |
| Most frequent chronic conditions, | n (%, 95% CI) |
| High blood pressure | 468 (78.9; 75.6–82.2) |
| Dyslipidemia | 301 (50.8; 46.7–54.8) |
| Diabetes | 250 (43.3; 38.1–46.1) |
| Osteoarthritis (knee, hand, hip and other) | 225 (37.9; 34.0–41.9) |
| Anxiety/Depression | 176 (29.7; 26.0–33.4) |
| Hypothyroidism | 113 (19.1; 15.9–22.2) |
| Obesity | 103 (17.4; 14.3–20.4) |
| Benign Prostatic Hyperplasia (% over Men) | 87 (33.2; 27.5–38.9) |
| Ischemic heart disease | 101 (17.0; 14.0–20.1) |
| Chronic heart failure | 21 (3.5; 2.0–5.0) |
| Asthma | 53 (8.9; 6.6–11.2) |
| Chronic Obstructive Pulmonary Disease | 64 (10.8; 8.3–13.3) |
| Osteoporosis | 75 (12.6; 10.0–15.3) |
| Atrial Fibrillation/Flutter | 73 (12.3; 9.7–15.0) |
| Esophageal diseases/peptic ulcers | 50 (8.4; 6.2–10.7)s |
| Number of drugs, median (IQR) | 7.0 (6.0, 9.0) |
| 5–6 drugs, n (%, CI) | 257 (43.3; 39.4–47.4) |
| 7–9 drugs, n (%, CI) | 230 (38.8; 34.9–42.8) |
| ≥10 drugs, n (%, CI) | 106 (17.9; 15.0–21.2) |
| **Physicians Characteristics (n = 117)** | |
| Age, M (SD) | 52.2 (6.8) |
| Age Range | 36–67 |
| Gender (women), n (%) | 77 (65.8) |
| Average length of physician career, M (SD) | 18.3 (3.4) |
| Postgraduate Medical Education Trainers, n (%) | 75 (64.1) |

*O'Halloran list of Chronic Conditions for ICPC

M: Mean. SD: Standard Deviation; IQR: Interquartile Range; CI: Confidence interval

**Table 2. Anatomical therapeutics classification groups of drugs prescribed.**

| ATC | Group description | N = 4,386 (%; 95% CI) |
|---|---|---|
| C Group | Cardiovascular system | 1,494 (34.1; 32.7–35.5) |
| A Group | Metabolism and alimentary tract | 952 (21.7; 20.5–23.0) |
| N Group | Nervous system | 879 (20.0; 18.9–21.3) |
| B Group | Blood and Blood forming organs | 392 (8.9; 8.1–9.8) |
| R Group | Respiratory system | 248 (5.7; 5.0–6.4) |
| H Group | Systemic hormonal preparations, excluding sex hormones and insulins | 142 (3.2; 2.8–3.9) |
| M Group | Musculo-skeletal system | 141 (3.2; 2.7–3.9) |
| G Group | Genito-urinary system and sex hormones | 112 (2.6; 2.1–3.1) |
| L Group | Antineoplastic and immunomodulating agents | 26 (0.6; 0.4–0.9) |

shows the distribution of inappropriate prescribing for each criterion. Omeprazole was the drug that accounted for the vast majority of inappropriate prescribing, having almost one in every five patients (19.9%) with an inappropriate MAI criterion rating. Additional information for the most common type of drug rated inappropriate can be found in S1 Table.

Based on the updated 2014 STOPP criteria, CPAS CheraktheMeds® detected PIP (at least one explicit criterion) for 340 patients (57.4%), 83 more than with the 2008 version (43.3%) (see Table 4). Eighty-two patients (13.8%) had 2 or more PIP with STOPP 2014 and 64 with STOPP 2008 (10.8%). The most frequently found PIP was the prolonged use of benzodiazepines (BZDs) for 217 patients (36.6%) using STOPP 2014 and 87 using STOPP 2008 (14.7%).

More than two out of every three study participants (68.8%) met at least one of the 2019 updated Beers criteria, twelve less than when assessing the participants with the 2015 version (70.8%). For these patients, 197 (33.2%)had two or more PIP using Beers 2015 and 193 (30.6%) using Beers 2019. Applying the 2019 Beers criteria, the most frequent PIP was the prolonged used of proton-pump inhibitors (PPIs) by 260 patients (43.8%). This PIP is also the most frequent using Beers 2015 (45.4%).

**Table 3. Distribution of inappropriate prescribing for each MAI criterion per drug prescribed or per patient.**

| Medication Appropriateness Index Criteria | Drugs with an inappropriate MAI criterion | Patients with an inappropriate MAI criterion in at least one medication |
|---|---|---|
| | n (%; CI 95%) | n (%; CI 95%) |
| Indication | 430 (9.8; 9.0–10.7) | 285 (48.4; 44.3–52.4) |
| Effectiveness | 724 (16.1; 15.4–17.3) | 385 (65.4; 61.5–69.2) |
| Correct Dosage | 622 (14.2; 13.2–15.2) | 358 (60.8; 56.8–64.7) |
| Correct Directions | 51 (1.2; 0.9–1.5) | 46 (7.8; 5.6–10.0) |
| Practical Directions | 553 (12.6; 11.7–13.6) | 325 (55.1; 51.1–59.2) |
| Potentially Drug-Drug Interaction | 928 (21.2; 20.0–22.4) | 373 (63.3; 59.4–67.2) |
| Drug-Disease/Condition interaction | 421 (9.6; 8.8–10.5) | 178 (30.2; 26.5–34.0) |
| Duplication | 509 (11.6; 10.7–12.6) | 140 (23.8; 20.3–27.2) |
| Duration | 775 (17.7; 16.6–18.8) | 403 (68.4; 64.7–72.2) |
| Cost-effectiveness | 979 (22.3; 21.1–23.6) | 427 (72.5; 68.9–76.1) |

*Medication Appropriateness Index

**Table 4. Sensitivity, specificity and measurements of agreements between different implicit and explicit criteria for appropriateness prescribing evaluation.**

|  | MAI* | STOPP 2014 | STOP 2008 | Beers 2019 | Beers 2015 |
|---|---|---|---|---|---|
| Prevalence of PIP (95% CI) | 94.1% (92.1–96) | 57.4% (53.7–61.7) | 43.3% (39.6–47.7) | 68.8% (65.1–72.5) | 70.8% (67.3–74.6) |
| Sensitivity (95% CI) | - | 60.1% (55.9–64.2) | 45.3% (41.1–49.6) | 68.8% (64.7–72.6) | 71.8% (67.9–75.6) |
| Specificity (95% CI) | - | 80% (63.1–91.6) | 82.9% (66.4–93.4) | 31.4% (16.9–49.3) | 42.9% (26.3–60.6) |
| Positive Predictive Value (95% CI) | - | 97.9% (95.8–99.2) | 97.7% (95–99.1) | 94.1% (91.3–96.2) | 95.2% (92.7–97.1) |
| ROC area (95% CI) | - | 0.7 (0.63;0.77) | 0.64 (0.57;0.71) | 0.5 (0.42;0.58) | 0.57 (0.49;0.66) |
| kappa index (p-value) (95% CI) | - | 0.104 (0.056;0.151) | 0.057 (0.025;0.089) | 0.001 (-0.052;0.054) | 0.052 (-0.009;0.113) |

*Medication Appropriateness Index

## Comparisons between the implicit criteria (gold standard) and updated explicit criteria

Table 4 shows the prevalence rates and the sensitivity and specificity for the two pair of versions of the explicit criteria comparing the old with the updated versions (STOPP 2008, STOPP 2014, Beers 2015 and Beers 2019) and comparing all of them with the implicit MAI criteria considered as the reference standard. A Venn diagram shows the agreement among the versions (Fig 1).

For the original version of the MAI, Beers 2015 had the highest sensitivity (71.8%) to detect PIP, followed by Beers 2019 (68.8%) and STOPP 2014 (60.1%). STOPP 2008 had the highest specificity (82.9%). The highest positive predictive value was obtained for STOPP 2014 (97.9%). The measurements of agreement (kappa indexes) were 0.104 between STOPP 2014 and the MAI, 0.057 between STOPP 2008 and the MAI and 0.001 between Beers 2019 and the MAI criteria.

There was no significant difference between the number of patients with PIP identified electronically by CPAS using Beers and STOPP criteria in their two more updated versions (p = 0.277). There were significant correlations between PIP identified by STOPP 2014 and the MAI (r = 0.192; p<0.001), STOPP 2008 and the MAI (r = 0.185; p<0.001) and Beers 2015 and the MAI (r = 0.311; p<0.001). The highest correlation was found between STOPP 2014 and

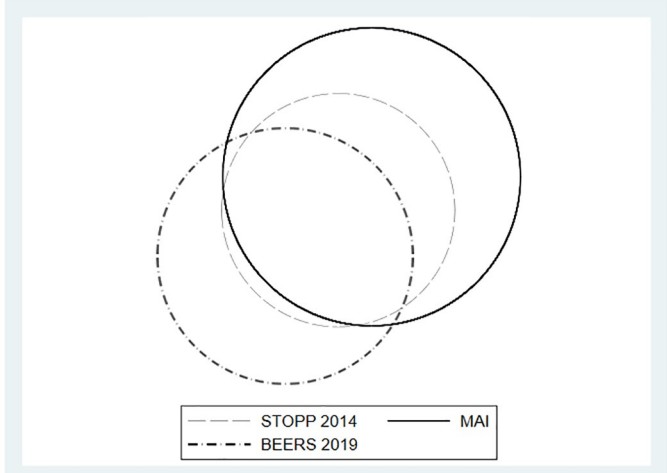

**Fig 1. Venn diagram.** Agreement among the versions.

**Table 5. Factors associated with summated medication appropriateness index scores.**

| INCREASING | Coefs. | CI 95% | p |
|---|---|---|---|
| Number of drugs | 2.41 | 1.46;3.35 | <0.001 |
| Usage of drugs in ATC A group (metabolism) | 4.83 | 2.02;7.64 | 0.001 |
| Usage of drugs in ATC M group (musculoskeletal) | 4.81 | 1.85;7.77 | 0.002 |
| Usage of drugs in ATC N group (nervous system) | 3.31 | 0.87;5.74 | 0.008 |
| DECREASING | | | |
| Diabetes | -6.29 | -9.00; -3.58 | <0.001 |
| Ischaemic Heart Disease | -6.92 | -10.61; -3.23 | <0.001 |
| Asthma | -9.13 | -12.59; -5.67 | <0.001 |
| Postgraduate Medical Education Trainers | -5.52 | -9.60; -1.44 | <0.001 |

PIP: Potentially inappropriate prescription; Coef: Coefficients; CI: Confidence interval

ATC: Anatomical Therapeutic Chemical classification

Beers 2015 (r = 0.358; p<0.001) (correlation in S2 Table; the distribution of summated MAI score per number of explicit criteria can be found in S3 Table).

## Factors independently associated with inappropriate prescribing based on implicit criteria and a comparison among explicit criteria methods

Based on the multivariable linear regression analysis for the summated MAI score, diabetes, ischaemic heart disease and asthma were independently associated with lower summated MAI scores (see Table 5). For every new drug taken by a patient, the MAI score increased by 2.41 (95% CI 1.46; 3.35) points. Out of all the physician factors studied, patients of doctors working also as postgraduate medical supervisors had lower MAI scores (coef. -5.52, 95% CI -9.60; -1.44).

Based on the multivariable logistic regressions performed for the MAI, STOPP and Beers criteria, taking more than 10 drugs was independently associated with inappropriate prescribing according to the MAI (OR: 20.86; 95% CI: 2.09;207.78, p = 0.010), STOPP 2014 criteria (OR: 4.96; 95% CI: 2.77;8.88, p<0.001), STOPP 2008 criteria (OR: 7.88; 95% CI: 4.48;13.8, p<0.001) and Beers 2015 criteria (OR: 3.31; 95% CI: 1.77;6.22, p<0.001). This association was not found for the Beers 2019 criteria. Some of the clinical conditions were found to be related with inappropriate medication use, such as anxiety or depression, across the four main criteria or the presence of osteoarthritis (see S4 Table). Fig 2 presents the estimated effects for different patient and physician characteristics for the two most sensitive explicit criteria compared to the MAI.

## Discussion

### Main findings

This study estimates, for the first time, the prevalence of PIP using the latest versions of explicit internationally accepted criteria and compares it with implicit criteria (MAI). The analysis of the criteria was performed by family doctors, with the help of a CPAS, for patients with multimorbidity and polypharmacy in primary care in Spain.

A total of 94.1% of the patients in this study had at least one criterion of inappropriateness for one of their drugs according to the MAI. PIP was detected in 57.7%, 43.6%, 68.8% and 71% of patients according to STOPP 2014, STOPP 2008, Beers 2019 and Beers 2015, respectively.

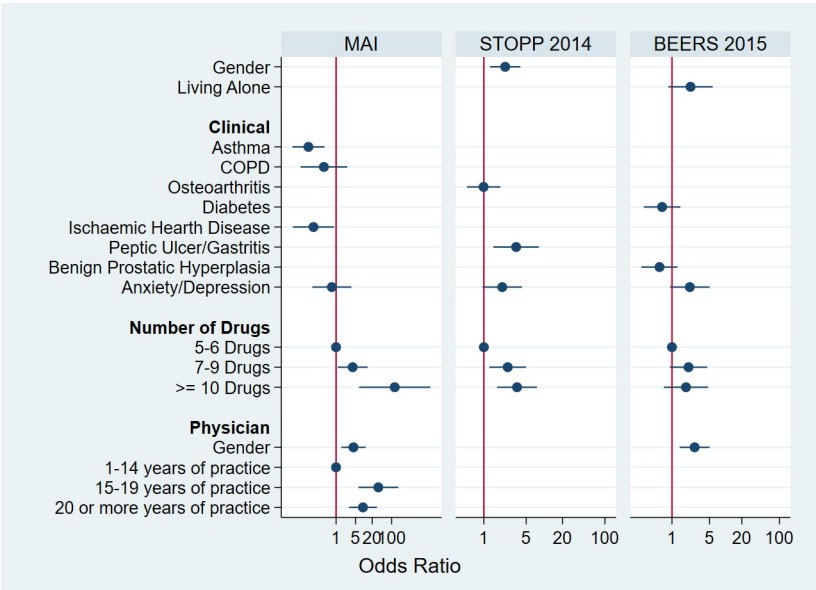

**Fig 2. Estimated effects for different patient and physician characteristics for the two most sensitive explicit criteria compared to the MAI.**

Patients with more than 10 drugs had a significantly greater presence of inappropriateness using any of the four criteria. The MAI detected greater inappropriateness than did the explicit criteria. Using the MAI as the gold standard, the criteria with the highest sensitivity was Beers 2015; however, the highest positive predictive value was obtained with STOPP 2014, with better concordance measures to detect PIP compared with other versions of STOPP or Beers.

## Prevalence of PIP

**Implicit criteria.**   The prevalence of PIP varies according to the criteria used and the characteristics of the population and studies [14,15,42]. The mean score of the summated MAI score per drug was 1.4 (2.3), and the mean MAI score per patient was 17.5 (16.8). A total of 94.1% of the patients presented an inappropriate MAI criterion for a drug. These results were similar to those reported for a study conducted in the primary care setting in the United States with polymedicated patients aged>65 years. This study detected PIP in 98.7% of patients and a summated MAI score per patient of 15.8 (10.1 SD) [17]. In another study in the primary care setting in Kuwait [21], the prevalence of PIP and mean MAI score were lower (PIP in 73.6% and MAI score per patient of 5.8 (5.8 SD)), but the population in this case had a mean consumption of drugs below that in our study. In a systematic review by Patterson et al. [43] of patients with polypharmacy aged over 65 years included in clinical trials from various areas, this score ranged from 6.5 to 19.3 in a total of 965 participants. In the PRIMUM trial [44], with a methodology and population similar to ours, the mean MAI was lower (4.6 (5.8 SD) in the control group and 4.8 (5.4 SD) in the intervention group). It is possible that the way in which the professional (pharmacist or doctor) evaluated patients with the MAI was not the same in the studies. The results will be different when the MAI is performed together with the patient, reviewing the clinical history (as in our study), or when only limited data are available from a

DCN. There are some limited data regarding the summated MAI score in Spanish primary care population. Only data from palliative care patients in Spain [45] or primary care in France [18,46].

The most frequent MAI individual criterion found in our study also differ from those found in the literature [17]. The item cost-effectiveness, the most frequent observed for our patients, could be biased due to the different prices of drugs in the different regions. This item has been eliminated in the adapted versions used in the studies that we previously indicated [21,44]. The criterion "potential drug-drug interactions" was much more frequent in our work than in the previous studies [21]. This difference can be related to the support tools for the detection of interactions used by the rater. In our case, a CPAS was used that provides information from various sources in an automated manner; this may have helped to better detect interactions.

Language variations and different MAI-criteria weighting reduced the comparability of these studies and can explain the differences found.

**Explicit criteria.**   Using the Beers criteria, the prevalence was 68.8% and 70.8% for the 2019 and 2015 versions, respectively. To date, there are no data available to compare the latest versions (2015 and 2019). For the 2015 version, the data obtained is slightly higher than those reported by previous studies in a hospital population (53.5%) [47], and similar to studies conducted with insurance databases (72.8%) [14,48]. With STOPP, this study obtained prevalence of 57.4% and 43.6% using STOPP 2014 and 2008, respectively, which is higher than those reported by prior studies [49,50] using previous STOPP versions in the primary care setting and in the European population (36% and 39%, respectively). In those who used the 2014 version, the prevalence ranged between 8.7 and 40.4% [16,41]; in a polymedicated geriatric primary care population in Kuwait, similar results were found (55.7%) [21]. The greater prevalence observed in this study could result from the included population, which had to meet the polypharmacy criterion to participate in the Multi-PAP trial, whereas only 72.9% or 72.1% of the patients in the abovementioned study were polymedicated.

Our results coincide with those of the majority of previous studies, placing BZDs and PPIs among the most frequently detected PIP with explicit criteria [14–16,41,49,51]. The percentage of patients in our sample who used BZDs for more than four weeks (STOPP v2) was similar to that obtained by Blanco Reina [13] (36.6% vs. 38.6%). The prolonged use of PPIs was the PIP most frequently detected in our sample, with both Beers versions (2019 (43.8%) and 2015 (45.5%)). The Beers 2015 update included this criterion, and there are already authors who describe the prolonged use of PPIs as frequent [47], with 41.9%; however, the samples are not comparable to ours.

## Comparison between implicit criteria and explicit criteria

Taking the MAI as the gold standard, Beers 2015 had the highest sensitivity (71.8%) to detect PIP, followed by Beers 2019 (68.8%) and STOPP 2014 (60.1%). The highest positive predictive value was obtained using STOPP 2014 (97.9%). Our data differ from those obtained in the Kuwait study [21] in terms of the sensitivities obtained in the explicit criteria. However, given the characteristics of our population and the high prevalence of PIP, STOPP 2014 could have a better diagnostic yield by having a greater positive predictive value. In other words, when faced with a positive STOPP 2014 value, it would be more likely that PIP would actually be confirmed by the MAI in patients with multimorbidity and polypharmacy. This is especially relevant in populations with *a priori* suspicion of a high prevalence of inappropriateness. This differs from similar studies in which the median number of drugs was lower and the population was different [21].

In Steinman's study they found that patients with a MAI score of three or more points on any drug had a Beers criterion of 34%. In hospitalized patients, those with a Beers criterion at admission had a mean of 7.17 (0.11–14.23) summated-MAI score [52].

We obtained a higher prevalence of PIP with the MAI. These implicit criteria are potentially the most sensitive ones and can take into account patients' preferences. However, these methods are time-consuming and it requires intensive training and extensive knowledge [8,20]. This means that they may not be the first option in clinical settings. Other tools, such as computer-automated explicit criteria, may be better options as an initial approach to determining the risk of drug inappropriateness.

It is evident that the common use measures of the quality of prescriptions produce widely discordant results (different quality metrics measuring different constructs). In studies comparing explicit methods with each other, the concordance among methods was low [16,51,53]. The different tools available can be complementary, and the decision to use one or the other should be related to the setting in which it is used and the purpose of its use. In research settings, a robust evaluation of the quality of prescriptions is probably necessary. However, in clinical practice, the use of explicit criteria, with less clinical detail and more easily automatable, can be applied more easily and quickly.

## Factors associated with PIP

In this study, we found that diseases such as diabetes, ischaemic heart disease and asthma were significantly associated with a lower summated MAI score. It is possible that patients with these pathologies, which are well defined and delimited, with frequent and protocolled visits, are subjected to more follow-ups at which there is opportunity to properly review medication and to easily detect situations of drug inappropriateness. Patients of doctors working both, as family physicians and postgraduate medical supervisors, had lower summated MAI score. We have not found other studies that have reported similar results. This could be related to the greater training required for those working as supervisors and also because of increased awareness in terms of reviewing treatments for polymedicated patients. Additionally, there is a clear association between a greater number of drugs and the summated MAI score. Taking more than 10 drugs is associated independently with greater detection of PIP with explicit criteria and with the MAI. This result continues to make the cut-off of 10 drugs a possible proxy for complex multimorbidity and an important indicator of risk.

## Conclusion

In conclusion, the prevalence of PIP detected in the sample was high and in agreement with previous literature in young senior population with multimorbidity and polypharmacy. The MAI criteria detected greater inappropriateness than did the explicit criteria, but their application was more complex and difficult to automate.

Future studies should evaluate whether the application of implicit and explicit criteria with the help of automated tools by a professional alone, among others, could change the usual clinical practice and improve patient health.

## Strengths and limitations

We found that this study covers an objective for which it was not initially designed. There was an attempt to partially correct this fact by estimating the aforementioned power as well as estimating the intervals with robust estimators to control the randomness introduced by the type of sampling. However, although the baseline information is trustworthy and was collected by each patient's doctor and verified, the MAI evaluation, which was performed by another

external physician, was not performed with the patient in an interview but rather by reviewing each patient's medical history, which may have resulted in an inappropriate valuation of some criteria related to the perspectives of the patient and the prescribing physician that may not be included in medical records.

Among the strengths, it is noteworthy that this is the first Spanish study that automatically evaluated PIP with Beers 2019. It is also the first one comparing explicit and implicit criteria in its latest versions with the MAI. In addition, data collection through an interview with each patient's own doctor, the pragmatic design and an exhaustive review of each patient's clinical history by the raters confer reliability to the data in this work.

Future studies should evaluate whether the application of implicit and explicit criteria with the help of automated tools by a professional alone, among others, could change the usual clinical practice and improve patient health.

## Supporting information

**S1 Table. Qualitative distribution of drug type and inadequacy in the 4 questions with the greatest relative weight in the medication appropriateness index of total prescriptions and patients.**
(XLSX)

**S2 Table. Correlation table.** Correlation between criteria.
(XLSX)

**S3 Table. Quantitative distribution of summated MAI score according to the number of explicit criteria detected.**
(XLSX)

**S4 Table. Factors associated with inappropriate prescribing as determined by MAI criteria, STOPP 2008 and 2014 versions and Beers 2015 and 2019 criteria.**
(XLSX)

## Acknowledgments

To our colleagues from the Research Unit, including the assistants Marcial Caboblanco Muñoz, and Juan Carlos Gil Moreno for their unconditional support. To all patients for their contribution to this research.

**MULTIPAP GROUP**:

**Lead authors for the MULTIPAP Study group**: Alexandra Prados-Torres (EpiChron Research Group, Aragon Health Sciences Institute (IACS), IIS Aragón, REDISSEC ISCIII, Miguel Servet University Hospital, Zaragoza, Spain) sprados.iacs@aragon.es, Juan Daniel Prados-Torres (Multiprofessional Teaching Unit for Family and Community Care Primary Care District Málaga-Guadarhorce, Instituto de Investigación Biomédica de Málaga–IBIMA-, Universidad de Málaga, Spain) juand.prados.sspa@juntadeandalucia.es, Isabel del Cura (Research unit. Primary Health Care Management Madrid. Spain) isabel.cura@salud.madrid.org.

**Coordinating committee**

María José Bujalance-Zafra (Victoria PCHC, Andalousian Health Service, Instituto de Investigación Biomédica de Málaga–IBIMA-, Universidad de Málaga, Spain), Francisca Leiva-Fernández (Multiprofessional Teaching Unit for Family and Community Care Primary Care District Málaga-Guadarhorce, Instituto de Investigación Biomédica de Málaga–IBIMA-, Universidad de Málaga, Spain), Fernando López Verde (Las Delicias PCHC, Andalousian Health Service, Málaga, Spain), Mercedes Aza-Pascual-Salcedo (EpiChron Research Group, Aragon

Health Sciences Institute (IACS), IIS Aragón, REDISSEC ISCIII. Aragon Health Service (SALUD), Zaragoza, Spain), Luis Gimeno-Feliu (EpiChron Research Group, Aragon Health Sciences Institute (IACS), IIS Aragón, REDISSEC ISCIII. San Pablo Health Centre, Aragon Health Service (SALUD), Zaragoza, Spain), Antonio Gimeno-Miguel (EpiChron Research Group, Aragon Health Sciences Institute (IACS), IIS Aragón, REDISSEC ISCIII, Miguel Servet University Hospital, Zaragoza, Spain), Francisca González Rubio (EpiChron Research Group, Aragon Health Sciences Institute (IACS), IIS Aragón, REDISSEC ISCIII. Delicias Sur Health Centre, Aragon Health Service (SALUD), Zaragoza, Spain), Victoria Pico-Soler (EpiChron Research Group, Aragon Health Sciences Institute (IACS), IIS Aragón, REDISSEC ISCIII. Torrero-LaPaz Health Centre, Aragon Health Service (SALUD), Zaragoza, Spain), Beatriz Poblador-Plou (EpiChron Research Group, Aragon Health Sciences Institute (IACS), IIS Aragón, REDISSEC ISCIII, Miguel Servet University Hospital, Zaragoza, Spain), Juan A López Rodríguez (Research unit. Primary Health Care Management Madrid. Spain), Cristina M Lozano Hernández (Research unit. Primary Health Care Management Madrid. Spain), Alessandra Marengoni (Department of Clinical and Experimental Sciences, University of Brescia, Brescia, Italy), Jesús Martín Fernández (Multiprofessional Teaching Unit for Family and Community Care Primary Care Oeste Primary Health Care Centre, Madrid, Spain), Christiane Muth (Institute of General Practice, Johann Wolfgang Goethe University, Frankfurt, Germany), Elena Polentinos Castro (Research unit. Primary Health Care Management Madrid. Spain), MA Elosisa Rogero (General Ricardos Primary Health Care Centre, Madrid, Spain), José María Valderas Martínez (University of Exeter Medical School, Exeter, UK. 22Department).

**Collaborating Investigators**

Maria del Pilar Barnestein-Fonseca (Fundación CUDECA, Instituto de Investigación Biomédica de Málaga–IBIMA-, Universidad de Málaga, Spain), (Marcos Castillo-Jimena (Coín PCHC, Malaga-Guadalhorce Sanitary District, Andalousian Health Service. Coín, Málaga, Instituto de Investigación Biomédica de Málaga (IBIMA), Universidad de Málaga, Spain), Miguel Domínguez-Santaella (Victoria PCHC, Malaga-Guadalhorce Sanitary District, Andalousian Health Service, Málaga, Spain), Nuria García-Agua-Soler (Department of Pharmacology, Faculty of Medicine, Malaga University), María Isabel Márquez-Chamizo (Carranque PCHC, Malaga-Guadalhorce Sanitary District, Andalousian Health Service, Málaga, Spain), José María Ruiz-San-Basilio (Coín PCHC, Malaga-Guadalhorce Sanitary District, Andalousian Health Service, Coín, Málaga, Spain), José María Abad-Díez (Department of Health, Social Welfare and Family, Government of Aragon), Marta Alcaraz Borrajo (Subdirectorate General of Pharmacy and Health Products), Gloria Ariza Cardiel (Multiprofessional Teaching Unit for Family and Community Care Primary Care Oeste Primary Health Care Centre, Madrid, Spain), Amaya Azcoaga Lorenzo (Pintores Primary Health Care Centre, Madrid, Spain), José María Abad-Díez (Aragon Health Service (SALUD), Government of Aragon, Zaragoza, Spain), Marta Alcaraz Borrajo (Subdirectorate General of Pharmacy and Health Products), Ana Cristina Bandrés-Liso (EpiChron Research Group, Aragon Health Sciences Institute (IACS), IIS Aragón, REDISSEC ISCIII. Aragon Health Service (SALUD), Zaragoza, Spain), Amaia Calderon-Larrañaga (EpiChron Research Group, Aragon Health Sciences Institute (IACS), IIS Aragón, REDISSEC ISCIII, Zaragoza, Spain. Aging Research Center, Department of Neurobiology, Care Sciences and Society, Karolinska Institutet, Stockholm University, Stockholm, Sweden.) Mercedes Clerencia-Sierra (EpiChron Research Group, Aragon Health Sciences Institute (IACS), IIS Aragón, REDISSEC ISCIII. Miguel Servet University Hospital, Aragon Health Service (SALUD), Zaragoza, Spain), Javier Marta-Moreno (EpiChron Research Group, Aragon Health Sciences Institute (IACS), IIS Aragón, REDISSEC ISCIII. Miguel Servet University Hospital, Aragon Health Service (SALUD), Zaragoza, Spain), Antonio Poncel-

Falcó (EpiChron Research Group, Aragon Health Sciences Institute (IACS), IIS Aragón, REDISSEC ISCIII. Aragon Health Service, Zaragoza, Spain), Ana Isabel González González (Technical Support Unit, Primary Care Management, Madrid Health Service), Virginia Hernández Santiago (Ninewells Hospital & Medical School, Dundee, UK), Angel Mataix SanJuan (Subdirección General de Farmacia y Productos Sanitarios), Ricardo Rodríguez Barrientos (Research unit. Primary Health Care Management Madrid. Spain), Mercedes Rumayor Zarzuelo (Centro de Salud Pública de Coslada, Área II Subdirección de Promoción de la Salud y Prevención), Luis Sánchez Perruca (Dirección Sistemas de Información, Gerencia Asistencial de Atención Primaria, Servicio Madrileño de Salud), Teresa Sanz Cuesta (Research unit. Primary Health Care Management Madrid. Spain), Mª Eugenia Tello Bernabé (El Naranjo PCHC, Madrid, Spain).

**Collaborating Investigators: Clinical Investigators in Primary Care Health Centres (PCHC)**:

**(Andalucía): PCHC Alhaurín el Grande** Javier Martín Izquierdo, Macarena Toro Sainz. **PCHC Carranque**: Mª José Fernández Jiménez, Esperanza Mora García, José Manuel Navarro Jiménez. **PCHC Ciudad Jardín**: Leovigildo Ginel Mendoza, Luz Pilar de la Mota Ybancos, Jaime Sasporte Genafo. **PCHC Coín**: Mª José Alcaide Rodríguez, Elena Barceló Garach, Beatriz Caffarena de Arteaga, Mª Dolores Gallego Parrilla, Catalina Sánchez Morales. **PCHC Delicias**: Mª del Mar Loubet Chasco, Irene Martínez Ríos, Elena Mateo Delgado. **PCHC La Roca**: Esther Martín Aurioles. **PCHC Limonar**: Sylvia Hazañas Ruiz. **PCHC Palmilla**: María Auxiliadora Nieves Muñoz Escalante. **PCHC Puerta Blanca**: Enrique Leonés Salido, Mª Antonia Máximo Torres, Mª Luisa Moya Rodríguez, María encarnación Peláez Gálvez, José Manuel Ramírez Torres, Cristóbal Trillo Fernández. **PCHC Tiro Pichón**: Mª Dolores García Martínez Cañavate, Mª del Mar Gil Mellado, Mª Victoria Muñoz Pradilla. **PCHC Vélez Sur**: Mª José Clavijo Peña, José Leiva Fernández, Virginia Castillo Romero. **PCHC Vera**: Rubén Vázquez Alarcón. **PCHC Victoria**: Rafael Ángel Maqueda, Gloria Aycart Valdés, Ana Mª Fernández Vargas, Irene García, Antonia González Rodríguez, Mª Carmen Molina Mendaño, Juana Morales Naranjo, Francisco Serrano Guerra.**PCHC Alcorisa** (Alcorisa): Carmen Sánchez Celaya del Pozo. **(Aragón): PCHC Alcorisa** (Alcorisa): Carmen Sánchez Celaya del Pozo. **PCHC Delicias Norte** (Zaragoza): José Ignacio Torrente Garrido, Concepción García Aranda, Marina Pinilla Lafuente, Mª Teresa Delgado Marroquín. **PCHC Picarral** (Zaragoza): Mª José Gracia Molina, Javier Cuartero Bernal, Mª Victoria Asín Martín, Susana García Domínguez. **PCHC Fuentes de Ebro** (Zaragoza): Carlos Bolea Gorbea.**PCHC Valdefierro** (Zaragoza): Antonio Luis Oto Negre. **PCHC Actur Norte** (Zaragoza): Eugenio Galve Royo, Mª Begoña Abadía Taira.**PCHC Alcañiz** (Alcañiz): José Fernando Tomás Gutiérrez. **PCHC Sagasta—Ruiseñores** (Zaragoza): José Porta Quintana, Valentina Martín Miguel, Esther Mateo de las Heras, Carmen Esteban Algora. **PCHC Ejea** (Ejea de los Caballeros): Mª Teresa Martín Nasarre de Letosa, Elena Gascón del Prim, Noelia Sorinas Delgado, Mª Rosario Sanjuan Cortés. **PCHC Canal Imperial—Venecia** (Zaragoza): Teodoro Corrales Sánchez. **PCHC Canal Imperial—San José Sur** (Zaragoza): Eustaquio Dendarieta Lucas. **PCHC Jaca** (Jaca): Mª del Pilar Mínguez Sorio. **PCHC Santo Grial** (Huesca): Adolfo Cajal Marzal. **(Madrid)PCHC Mendiguchía Carriche** (Leganés): Eduardo Díaz García, Juan Carlos García Álvarez, Francisca García De Blas González, Cristina Guisado Pérez, Alberto López García Franco, Mª Elisa Viñuela Benitez. **PCHC El Greco** (Getafe): Ana Ballarín González, Mª Isabel Ferrer Zapata, Esther Gómez Suarez, Fernanda Morales Ortiz, Lourdes Carolina Peláez Laguno, José Luis Quintana Gómez, Enrique Revilla Pascual. **PCHC Cuzco** (Fuenlabrada): M Ángeles Miguel Abanto.**PCHC El Soto** (Móstoles): Blanca Gutiérrez Teira. **PCHC General Ricardos** (Madrid): Francisco Ramón Abellán López, Carlos Casado Álvaro, Paulino Cubero González, Santiago Manuel Machín Hamalainen, Raquel Mateo Fernández, Mª Eloisa Rogero Blanco,

Cesar Sánchez Arce. **PCHC Ibiza** (Madrid): Jorge Olmedo Galindo. **PCHC Las Américas** (Parla): Claudia López Marcos, Soledad Lorenzo Borda, Juan Carlos Moreno Fernández, Belén Muñoz Gómez, Enrique Rodríguez De Mingo. **PCHC Mª Ángeles López** (Leganés): Juan Pedro Calvo Pascual, Margarita Gómez Barroso, Beatriz López Serrano, Mª Paloma Morso Peláez, Julio Sánchez Salvador, Jeannet Dolores Sánchez Yépez, Ana Sosa Alonso. **PCHC Mª Jesús Hereza** (Leganés): Mª del Mar Álvarez Villalba. **PCHC Pavones** (Madrid): Purificación Magán Tapia. **PCHC Pedro Laín Entralgo** (Alcorcón): Mª Angelica Fajardo Alcántara, Mª Canto De Hoyos Alonso, Mª Aránzazu Murciano Antón. **PCHC Pintores** (Parla): Manuel Antonio Alonso Pérez, Ricardo De Felipe Medina, Amaya Nuria López Laguna, Eva Martínez Cid De Rivera, Iliana Serrano Flores, Mª Jesús Sousa Rodríguez. **PCHC Ramón y Cajal** (Alcorcón): Mª Soledad Núñez Isabel, Jesús Mª Redondo Sánchez, Pedro Sánchez Llanos, Lourdes Visedo Campillo.

## Author Contributions

**Conceptualization:** Juan A. Lopez-Rodriguez, Eloísa Rogero-Blanco, Francisca Leiva-Fernandez, J. Daniel Prados-Torres, Alexandra Prados-Torres.

**Data curation:** Juan A. Lopez-Rodriguez.

**Formal analysis:** Juan A. Lopez-Rodriguez, Eloísa Rogero-Blanco.

**Funding acquisition:** J. Daniel Prados-Torres, Alexandra Prados-Torres, Isabel Cura-González.

**Investigation:** Eloísa Rogero-Blanco, Mercedes Aza-Pascual-Salcedo, Fernando Lopez-Verde, Victoria Pico-Soler, Francisca Leiva-Fernandez.

**Methodology:** Juan A. Lopez-Rodriguez, Eloísa Rogero-Blanco, Isabel Cura-González.

**Supervision:** J. Daniel Prados-Torres, Alexandra Prados-Torres, Isabel Cura-González.

**Validation:** Juan A. Lopez-Rodriguez, Eloísa Rogero-Blanco, Mercedes Aza-Pascual-Salcedo, Fernando Lopez-Verde, Victoria Pico-Soler.

**Writing – original draft:** Juan A. Lopez-Rodriguez, Eloísa Rogero-Blanco, Isabel Cura-González.

**Writing – review & editing:** Juan A. Lopez-Rodriguez, Eloísa Rogero-Blanco, Mercedes Aza-Pascual-Salcedo, Fernando Lopez-Verde, Victoria Pico-Soler, Francisca Leiva-Fernandez, J. Daniel Prados-Torres, Alexandra Prados-Torres, Isabel Cura-González.

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
