## [Decision Letter · Decision Letter 0]

16 Mar 2020

PONE-D-20-04779

Potentially Inappropriate prescriptions according to explicit and implicit criteria in patients with multimorbidity and polypharmacy - MULTIPAP study: A cross-sectional

PLOS ONE

Dear MRs ROGERO,

Thank you for submitting your manuscript to PLOS ONE. After careful consideration, we feel that it has merit but does not fully meet PLOS ONE’s publication criteria as it currently stands. Therefore, we invite you to submit a revised version of the manuscript that addresses the points raised during the review process.

We would appreciate receiving your revised manuscript by Apr 30 2020 11:59PM. To enhance the reproducibility of your results, we recommend that if applicable you deposit your laboratory protocols in protocols.io, where a protocol can be assigned its own identifier (DOI) such that it can be cited independently in the future. For instructions see: http://journals.plos.org/plosone/s/submission-guidelines#loc-laboratory-protocols

We look forward to receiving your revised manuscript.

Kind regards,

Prof, Mojtaba Vaismoradi, PhD, MScN, BScN

Academic Editor

PLOS ONE

Journal Requirements:

3. One of the noted authors is a group or consortium MULTIPAP group. In addition to naming the author group, please list the individual authors and affiliations within this group in the acknowledgments section of your manuscript. Please also indicate clearly a lead author for this group along with a contact email address.

4. Your ethics statement must appear in the Methods section of your manuscript. If your ethics statement is written in any section besides the Methods, please move it to the Methods section and delete it from any other section. Please also ensure that your ethics statement is included in your manuscript, as the ethics section of your online submission will not be published alongside your manuscript.

Reviewers' comments:

Reviewer #1: Thank you for the opportunity to review this research work about the potentially inappropriate prescriptions in patients with multimorbidity and polypharmacy. I hope the following comments/ suggestions be useful in the development of this document.

I want to suggest it is needed to review the entire document by a native translator. The document needs changes that increase clarity and facilitate its reading.

Title: Second part of the title could be more precise. Please, consider: “Potentially Inappropriate prescriptions according to explicit and implicit criteria in patients with multimorbidity and polypharmacy. MULTIPAP: A cross-sectional study.”

Abstract: According to the “Submission Guidelines” the Abstract should not include citations and abbreviations, if possible. Please, check it.

- Line 44: Multi-PAP is written in capitals in the title. Please, you should unify it.

- Line 51: if you decide to include these cites, please, check how doing it.

Introduction: The contents of the introduction need to be related and exposed following a somewhat brighter argument line.

- Line 66-67: please include the cite/s for this definition.

- Lines 76, 97: the number must be before the "."

- Line 96: Change “In our country…” by “In Spain…”

- Line 97-98: Consider strengthening the rationale for this statement.

- Line 99: Rewrite in a proper way mean and standard deviation.

- Lines 99 to 106: Please, consider to examine this paragraph. It seems not clear enough. Something is missing in the last sentence.

- Line 110, 112: Review the cite.

- Lines 114-116: Please, consider to examine this paragraph. It seems not clear enough.

- Line 124: Compare and concur the aim of the abstract and the introduction.

- Lines 125-126: Review the cites.

Material and methods:

- Line 148: the data were collected December 2016 – January 2017? Have authors thought about include data from 2019 or 2020?

- There were used the same method of data collection in patients and physicians? Please, clarify the method and variables for each group of participants.

Results

- What is the range age of the patients?

- Line 236: It is explained that “digestive system/metabolism” is one of the most popular group prescriptions, but in table 2, this group is identified as “metabolism and alimentary tract”. Please, check and change this issue.

Discussion

-Line 349: The authors affirm the absence of studies with MAI quantitative data based on a 2013 article. Please check that the absence of evidence exists currently.

The conclusion section is missing.

Tables: check them and give the format according to the “Submission Guidelines”.

Table 1: Complete with additional results (age range for patients & physicians, male gender, …)

Table 3: Consider to remove column 1.

References: Check and make changes following the “Submission Guidelines”.

Reviewer #2: General comments

- Pay attention to punctuation for instance in lines 76, 97.

Introduction

- Lines 67-68, can the authors point to a reference to back up it?

Method

- Why did you choose this inclusion criterion for the age of 74, for your study?

- Delete line 141, “Participating physicians did not receive financial compensation”.

- Line 143: “from three Spanish regions …”, how were these regions selected? Based on which method?

- Line 155: First time an abbreviation is used, the term should be spelled out in full.

Results

- This section needs minor rectifications. Please consult with a statistician for how the section should be journalistic way written (lines 223-131).

Discussion

- Line 343: delete (5 studies).

- Lines 351 (… differ from those found in the literature.), 355 (… than in the previous studies): can the authors point to some reference to back up them?

Limitations

- Line 427, delete (Among the main limitations) and then add all limitations of your study.

Also, please add “Conclusion” section and mention main findings of your study, implications.

---

## [Author Response · Author response to Decision Letter 0]

21 May 2020

Response to reviewers

Reviewer #1: Thank you for the opportunity to review this research work about the potentially inappropriate prescriptions in patients with multimorbidity and polypharmacy. I hope the following comments/ suggestions be useful in the development of this document. 

I want to suggest it is needed to review the entire document by a native translator. The document needs changes that increase clarity and facilitate its reading.

Thank you very much for the suggestion regarding the language. We tried our best by translating and editing it by American Journal Expert. We attached the translation certification. We considered the American native speaker for the scope of this journal. We re-reviewed the translation and incorporated some changes to improve readability. Your suggestions have also been sent to AJE for clarification. 

Title: Second part of the title could be more precise. Please, consider: “Potentially Inappropriate prescriptions according to explicit and implicit criteria in patients with multimorbidity and polypharmacy. MULTIPAP: A cross-sectional study.”

We have incorporated your proposed title

Abstract: According to the “Submission Guidelines” the Abstract should not include citations and abbreviations, if possible. Please, check it.

- Line 44: Multi-PAP is written in capitals in the title. Please, you should unify it. MULTIPAP has been corrected and unified. 

- Line 51: if you decide to include these cites, please, check how doing it.

All abbreviations have been checked and revised to include. The abbreviation Potentially Inappropriate Prescription (PIP) has been removed from the abstract. The Medication Appropriateness Index (MAI) has been maintained as it is the most common name for the questionnaire, as well as the STOPP criteria.

Introduction: The contents of the introduction need to be related and exposed following a somewhat brighter argument line.

- Line 66-67: please include the cite/s for this definition. The reference has been included.

- Lines 76, 97: the number must be before the "." The number has been placed before the [ ]

- Line 96: Change “In our country…” by “In Spain…” The change has been made.

- Line 97-98: Consider strengthening the rationale for this statement. It has been deleted to simplify since it doesn’t follow the narrative structure of the paragraph. 

- Line 99: Rewrite in a proper way mean and standard deviation. The change has been made.

- Lines 99 to 106: Please, consider to examine this paragraph. It seems not clear enough. Something is missing in the last sentence. The paragraph has been revised and the acronym PC has been replaced throughout the text by primary care to make it more reader-friendly.

- Line 110, 112: Review the cite. The cite has been revised.

- Lines 114-116: Please, consider to examine this paragraph. It seems not clear enough. Paragraph has been restated to help understanding. 

- Line 124: Compare and concur the aim of the abstract and the introduction. The aims have been compared and their wording unified.

- Lines 125-126: Review the cites. Lines 125 and 126 correspond to the wording of the objective, they do not contain cites.

Material and methods:

- Line 148: the data were collected December 2016 – January 2017? Have authors thought about include data from 2019 or 2020?

Sociodemographic and clinical data were obtained during the period of patient recruitment in the MULTIPAP study, which was conducted between December 2016 and January 2017 through an interview with each patient's family doctor. It is the baseline data of the study that are the object of interest of this work.

- There were used the same method of data collection in patients and physicians? Please, clarify the method and variables for each group of participants.

The method of data collection on patients and physicians was not the same.

The method of collecting data on patients and doctors was not the same. Data on professionals (age, sex, length of physician career, and being postgraduate medical supervisor) were recorded by each physician in the data collection notebook after signing their commitment to collaborate in the study. Subsequently, sociodemographic and clinical data on patients were collected through an interview with each patient's general practitioner after signing the informed consent.

Results

- What is the range age of the patients?

Since inclusion criteria were strict trying to look for “young seniors”, the age range is exactly the same as the inclusion criteria, being 65-74 years old. For this reason we didn’t include in the manuscript. Interquartile range was 68-72.

- Line 236: It is explained that “digestive system/metabolism” is one of the most popular group prescriptions, but in table 2, this group is identified as “metabolism and alimentary tract”. Please, check and change this issue.

The change has been made

Discussion

-Line 349: The authors affirm the absence of studies with MAI quantitative data based on a 2013 article. Please check that the absence of evidence exists currently.

We’ve reviewed the text and changed it to reflect what happens in the Spanish population. We included a more updated reference regarding the most alike population (French study by Gilbert). Citations has been included and updated. 

The conclusion section is missing.

Following the sections proposed by the journal Plos one, Conclusions (optional) the section has not been included.

Following the reviewer's proposal, we have incorporated an epigraph “Implications of the study findings” synthesizing the conclusions. If you consider that a specific section of conclusions is more appropriate, we will proceed to do so. 

Tables: check them and give the format according to the “Submission Guidelines”. Reviewed

Table 1: Complete with additional results (age range for patients & physicians, male gender, …) Included in the tables

Table 3: Consider to remove column 1.

We have removed column 1. 

References: Check and make changes following the “Submission Guidelines”. Cheked

 

Reviewer #2: General comments

Pay attention to punctuation for instance in lines 76, 97.The number has been placed before the [ ]

Introduction

- Lines 67-68, can the authors point to a reference to back up it?. It has been pointed out since it’s a shared comment with reviewer 1

Method

- Why did you choose this inclusion criterion for the age of 74, for your study?

In Spain, in the context of the Strategy of the Ministry of Health, Social Policy and Equality to improve the care of polymedicated chronic patients, programs aimed at polymedicated elderly patients (>75 years) have been implemented both nationally and in various regions. However, patients over 75 years old are only a part of the population with multi-morbidity. Patients who are sociologically define as young seniors (65-74 years old) have a high prevalence of multimorbility and polymedication and, although evidence on the effectiveness of interventions on this population group is even more limited, this is an age group with an important potential for early intervention. For this reason our study group focuses on this age range. 

- Delete line 141, “Participating physicians did not receive financial compensation”. The line has been deleted

- Line 143: “from three Spanish regions …”, how were these regions selected? Based on which method?

The three regions are Andalusia, Aragon and Madrid. They represent three regions geographically located in the south, centre and north of the country respectively. In each one, the Cooperative Research Network on Services and Chronic Diseases (REDISSEC) of the Carlos III Health Institute www.redisecc.com has a research group with proven research experience on patients with multimorbility and chronic diseases. Our research approach both, the epidemiological perspective of the characterization of chronicity and the evaluation of the effectiveness of interventions in patients with chronic diseases in primary care.

These groups from the three regions obtained funding in competitive public calls for the MULTIPAP study, which includes cross-sectional studies such as the one proposed here, clinical trials, and cohort studies. www.multipap.es

- Line 155: First time an abbreviation is used, the term should be spelled out in full. The term has been included in its complete form. The Organisation for Economic Co-operation and Development (OECD)

Results

- This section needs minor rectifications. Please consult with a statistician for how the section should be journalistic way written (lines 223-131).

Several members of the research team are statisticians. We understand that you are referring to the wording of the paragraph in the lines (lines 223-231). Following your suggestions we have simplified the wording of the paragraph.

Discussion

- Line 343: delete (5 studies). The words have been deleted

- Lines 351 (… differ from those found in the literature.), 355 (… than in the previous studies): can the authors point to some reference to back up them?

The three citations supporting those statements have been relocated according to the specific section. 

Limitations

- Line 427, delete (Among the main limitations) and then add all limitations of your study.

“Among the main limitations” has been deleted

Also, please add “Conclusion” section and mention main findings of your study, implications.

Following the sections proposed by the journal Plos one, Conclusions (optional) the section has not been included.

Following the reviewer's proposal, we have incorporated an epigraph “Implications of the study findings” synthesizing the conclusions. If you consider that a specific section of conclusions is more appropriate, we will proceed to do so.

---

## [Decision Letter · Decision Letter 1]

16 Jun 2020

PONE-D-20-04779R1

Potentially Inappropriate prescriptions according to explicit and implicit criteria in patients with multimorbidity and polypharmacy. MULTIPAP: A cross-sectional study

PLOS ONE

Dear Dr. ROGERO,

Thank you for submitting your manuscript to PLOS ONE. After careful consideration, we feel that it has merit but does not fully meet PLOS ONE’s publication criteria as it currently stands. Therefore, we invite you to submit a revised version of the manuscript that addresses the points raised during the review process.

We look forward to receiving your revised manuscript.

Kind regards,

Prof, Mojtaba Vaismoradi, PhD, MScN, BScN

Academic Editor

PLOS ONE

Reviewers' comments:

Reviewer #1: Dear author thanks for increasing manuscript quality with your efforts. Nevertheless, some of the changes suggested are missing yet.

Line 39-41: sample size has been removed from the abstract. Please, add it again.

Line 51-52: Add the bibliographic reference of each criterion: STOPP 2014, STOPP 2008, Beers 2019 & Beers 2015.

Introduction: I suggested in previous inform: "The contents of the introduction need to be related and exposed following a somewhat brighter argument line" and nothing has been done with this issue.

In response to reviewers, it is answered to my question about the method of data collection in patients and physicians. Please, include in the manuscript, in the same manner, the explanation given to reviewers.

Also, I would like to request again that be included the range age of patients in the results section, despite you explained it was an inclusion criteria.

Reviewer 2 and 1 agree with a conclusion section is needed in the manuscript. Please, consider to include it.

Reviewer #2: The only challenge to the paper is that it needs a bit of editing for use of English grammar. With that editing I recommend publication.

Table 1: Substitute “Female, Male” instead of “Women, Men”.

Lines 408-410: Grammar needs to be reviewed.

---

## [Author Response · Author response to Decision Letter 1]

22 Jun 2020

Response to reviewers

Reviewer #1: Dear author thanks for increasing manuscript quality with your efforts. Nevertheless, some of the changes suggested are missing yet.

Line 39-41: sample size has been removed from the abstract. Please, add it again. 

Sample size has been included

Line 51-52: Add the bibliographic reference of each criterion: STOPP 2014, STOPP 2008, Beers 2019 & Beers 2015.

After reviewing journal style, Abstract section, along with other sections, does not accept bibliographic references. We understood, and after an editor consultation, that what was asked was to include acronyms definition for the abbreviations included in the abstract. 

Introduction: I suggested in previous inform: "The contents of the introduction need to be related and exposed following a somewhat brighter argument line" and nothing has been done with this issue. 

The argument line in the introduction has been changed

In response to reviewers, it is answered to my question about the method of data collection in patients and physicians. Please, include in the manuscript, in the same manner, the explanation given to reviewers.

The explanation given to the reviewers previously has been included in methods section.

Also, I would like to request again that be included the range age of patients in the results section, despite you explained it was an inclusion criteria.

The age range was already included in Table 1 and has been moved to results section as suggested. 

Reviewer 2 and 1 agree with a conclusion section is needed in the manuscript. Please, consider to include it.

We have included the conclusions section

Reviewer #2: The only challenge to the paper is that it needs a bit of editing for use of English grammar. With that editing I recommend publication.

Table 1: Substitute “Female, Male” instead of “Women, Men”. 

The change has been made

Lines 408-410: Grammar needs to be reviewed.

We have reviewed the grammar in these lines

---

## [Decision Letter · Decision Letter 2]

15 Jul 2020

PONE-D-20-04779R2

Potentially Inappropriate prescriptions according to explicit and implicit criteria in patients with multimorbidity and polypharmacy. MULTIPAP: A cross-sectional study

PLOS ONE

Dear Dr. ROGERO,

Thank you for submitting your manuscript to PLOS ONE. After careful consideration, we feel that it has merit but does not fully meet PLOS ONE’s publication criteria as it currently stands. Therefore, we invite you to submit a revised version of the manuscript that addresses the points raised during the review process.

We look forward to receiving your revised manuscript.

Kind regards,

Prof, Mojtaba Vaismoradi, PhD, MScN, BScN

Academic Editor

PLOS ONE

Reviewers' comments:

Reviewer #1: Dear author/s thanks for your efforts to increase manuscript quality. Now Introduction section's ideas are ordered following a line of argument that will enjoy the PLOS ONE readers. Thanks for taking into account all my last recommendations.

Nevertheless, a very minor change is needed.

Lines 40-42- In order to increase clarity, this is my suggestion: "[...] according to American Geriatrics Society Beers Criteria® (2015, 2019), the Screening Tool of Older Person’s Prescription -STOPP- criteria (2008, 2014), and the Medication Appropriateness Index -MAI- criteria."

---

## [Author Response · Author response to Decision Letter 2]

17 Jul 2020

Response to reviewers

Reviewer #1:

Lines 40-42- In order to increase clarity, this is my suggestion: "[...] according to American Geriatrics Society Beers Criteria® (2015, 2019), the Screening Tool of Older Person’s Prescription -STOPP- criteria (2008, 2014), and the Medication Appropriateness Index -MAI- criteria 

The change has been made. It is true that it is clearer

---

## [Editor Report · Decision Letter 3]

22 Jul 2020

Potentially Inappropriate prescriptions according to explicit and implicit criteria in patients with multimorbidity and polypharmacy. MULTIPAP: A cross-sectional study

PONE-D-20-04779R3

Dear Dr. ROGERO,

We’re pleased to inform you that your manuscript has been judged scientifically suitable for publication and will be formally accepted for publication once it meets all outstanding technical requirements.

Kind regards,

Prof, Mojtaba Vaismoradi, PhD, MScN, BScN

Academic Editor

PLOS ONE

---

## [Editor Report · Acceptance letter]

24 Jul 2020

PONE-D-20-04779R3 

Potentially Inappropriate prescriptions according to explicit and implicit criteria in patients with multimorbidity and polypharmacy. MULTIPAP: A cross-sectional study 

Dear Dr. Rogero-Blanco:

I'm pleased to inform you that your manuscript has been deemed suitable for publication in PLOS ONE. Congratulations! Your manuscript is now with our production department. 

Kind regards, 

on behalf of

Professor Mojtaba Vaismoradi 

Academic Editor

PLOS ONE